



# Roll vortices induce new particle formation bursts in the planetary boundary layer

Janne Lampilahti[1], Hanna Elina Manninen[2], Katri Leino[1], Riikka Väänänen[1], Antti Manninen[3], Stephany Buenrostro Mazon[1], Tuomo Nieminen[1], Matti Leskinen[1], Joonas Enroth[1], Marja Bister[1], Sergej Zilitinkevich[1,3,4], Juha Kangasluoma[1,5], Heikki Järvinen[1], Veli-Matti Kerminen[1], Tuukka Petäjä[1,6], Markku Kulmala[1,5,6]

[1]Institute for Atmospheric and Earth System Research/Physics, Faculty of Science, University of Helsinki, Helsinki, Finland.
[2]CERN, CH-1211 Geneva, Switzerland.
[3]Finnish Meteorological Institute, Helsinki, Finland.
[4]Department of Radio-physics, University of Nizhny, Novgorod, Russia.
[5]Aerosol and Haze Laboratory, Beijing Advanced Innovation Center for Soft Matter Science and Engineering, Beijing University of Chemical Technology, Beijing, China.
[6]Joint International Research Laboratory of Atmospheric and Earth System Sciences, Nanjing University, Nanjing, China.

*Correspondence to*: Janne Lampilahti (janne.lampilahti@helsinki.fi)

**Abstract.** Recent studies have shown the importance of new particle formation (NPF) to global cloud concensation nuclei (CCN) production, as well as to air pollution in megacities. In addition to the necessary presence of low-volatility vapors that can form the new aerosol particles, both numerical and observational studies have shown that the dynamics of the planetary boundary layer (BL) plays an important role in NPF. Evidence from field observations suggests that roll vortices might be favorable for inducing NPF in a convective BL. However, direct observations and estimates on the potential importance of this phenomenon to the production of new aerosol particles are lacking. Here we show that rolls frequently induce NPF bursts along the horizontal circulations, and that the small clusters and particles originating from these bursts grow in size similar to particles typically ascribed to regional-scale atmospheric NPF. We outline a method to identify roll-induced NPF from measurements and, based on the collected data, estimate the impact of roll vortices on the overall aerosol particle production due to NPF at a boreal forest site (83±34% and 26±8% overall enhancement in particle formation for 3-nm and 10-nm particles respectively). We conclude that the formation of roll vortices should be taken into account when estimating particle number budgets in the atmospheric BL.



## 1 Introduction

Atmospheric new particle formation (NPF) is a globally important source of aerosol particles and cloud condensation nuclei (CCN) (Dunne et al., 2016; Gordon et al., 2017; Kerminen et al., 2018; Kulmala et al., 2004), having potentially large influences on climate via aerosol-cloud interactions (Boucher et al., 2013) as well as on human health by increasing ultrafine particle number concentrations. Numerical investigations have linked fluctuations in the ambient temperature and relative humidity, caused by for example small-scale turbulence, large eddies such as roll vortices, or mixing over a temperature inversion to significant enhancements in new particle formation rate compared to only mean conditions (Easter and Peters, 1994; Nilsson and Kulmala, 1998). In observational studies enhanced nucleation mode particle concentrations have been observed in turbulent layers in the lower atmosphere. For example inside the residual layer (Wehner et al., 2010) and in the inversion capping a shallow mixed layer (Platis et al., 2015; Siebert et al., 2004). Other airborne measurements have found significant horizontal and vertical variability in the number concentration of nucleation mode particles within the BL (Crumeyrolle et al., 2010; Leino et al., 2019; O'Dowd et al., 2009; Schobesberger et al., 2013; Väänänen et al., 2016). One possible reason for this could be the effect of bounadry layer (BL) dynamics.

Convection in the planetary BL often organizes into counter-rotating horizontal roll vortices or rolls that extend to the top of the boundary layer (Atkinson and Wu Zhang, 1996; Etling and Brown, 1993; Young et al., 2002). Buzorius et al. (2001) and Nilsson et al. (2001) noted that roll vortices commonly occurred during NPF events and suggested that they might be especially inducive to NPF (Buzorius et al., 2001; Nilsson et al., 2001).

However direct observations of roll vortices inducing NPF are lacking, and the overall effect of rolls on aerosol particle formation is unknown. In this study we have analyzed co-located airborne and ground-based measurements from southern Finland during 2013-2015 in order to determine the effect of roll vortices on NPF.



## 2 Methods

### 2.1 Zeppelin measurements.

In May-June 2013, in the framework of the PEGASOS (Pan-European Gas-AeroSOls Climate Interaction Study) project, aerosol particle and gas phase measurements were performed over Hyytiälä and Jämi in southern Finland using an instrumented Zeppelin NT (Neue Technologie) airship.

Here we analyzed measurements from the onboard Neutral cluster and Air Ion Spectrometer (NAIS) (Mirme et al., 2010; Mirme and Mirme, 2013) on May 8, 2013. The NAIS can measure the particle number-size distribution in the mobility diameter range 2-42 nm and ion number-size distribution in the mobility diameter range 0.8-42 nm. We used the positively charged particles and the data was averaged to 4 min time resolution.

During the measurement the inlet of the NAIS was pushed out from the window of the zeppelin's gondola. The data was corrected for diffusional losses in the one meter long, 37 mm inner diameter, inlet tube and converted to standard conditions (293.15 K and 1 atm). The temperature and pressure recorded by the instrument were used in the corrections. Any losses occurring at the inlet nozzle were assumed to be negligible due small size of the measured particles and relatively low airspeed, so that the particles closely followed streamlines.

The measurement profiles had slow ascends (~25 min) up to 1 km height above ground and fast descends (~5 min) while the airspeed was kept at ~20 m/s. The profiles were flown over the same circular area that was only ~4 km in diameter (see Figure 1). The flights started and ended at the Jämi airfield (61°46′43″N, 22°42′58″E, 154 m above sea level).

### 2.2 Airplane measurements

The University of Helsinki has organized several airborne measurement campaigns around Hyytiälä using an instrumented Cessna 172 airplane. Descriptions of the measurement setups can be found in previous works (Leino et al., 2019; Schobesberger et al., 2013; Väänänen et al., 2016). Table 1 shows a



summary of the airborne measurement campaigns and the instrumentation from which data was used in this study.

Particle number concentration in the 3-20 nm range was calculated by subtracting the total particle
number concentration measured by the Scanning Mobility Particle Sizer (SMPS) from the number concentration measured by the Ultrafine Condensation Particle Counter (UCPC). The SMPS starts to lose accuracy in terms of spatial distribution of the aerosol particles due to its 2 min averaging period when the horizontal scale becomes less than 4 km. A turbulence probe, capable of measuring the 3d wind vector, was only installed at the end of the 2015 campaign.

Typical measurement tracks consisted of ~30 km long flight segments flown roughly perpendicular to the mean wind direction over the same area such that the aircraft was either descending, ascending or staying level. The altitude range was between 100-3000 m above ground. The measurement airspeed was 36 m/s. Usually two 2.5 h flights were flown during a single day, one in the morning and one in the
afternoon. Vertically the measurements were able to probe all parts of the BL as well the lowest kilometer of the free troposphere. The flights started and ended at the Tampere-Pirkkala airport (61°24′55″N, 23°35′16″E, 119 m above sea level).

## 2.3 Ground-based measurements

The airborne measurements were complemented by the measurements at the SMEAR II field station.
The measurement station is located in Hyytiälä, Finland (61°50'40"N, 24°17'13"E, 180 m above sea level) and is surrounded by flat terrain and coniferous forest. The station represents the background conditions found in the boreal forest regions of northern latitudes (Hari and Kulmala, 2005).

The key aerosol instruments included in this study were the station's Differential Mobility Particle Sizer
(DMPS) (Aalto et al., 2001), the NAIS (Manninen et al., 2009) and the Particle Size Magnifier (PSM) (Vanhanen et al., 2011). From the NAIS the positively charged particles were used and the data was averaged to 4 min time resolution. The PSM measured particle number-size distribution between 1-2

nm and the data was averaged to 12 min time resolution. The DMPS sampled the air from a vertical inlet at 8 m above the ground and the NAIS through a wall inlet at 2 m above the ground, both were inside the canopy. The PSM was sampling in a 35 m tall tower, above the forest canopy. The aerosol particle data from the station was not converted to standard conditions since the correction would be negligible.

Measurements of meteorological variables (temperature, pressure, relative humidity, wind direction and speed) and vertical particle flux from the station's mast were available at 30 min time resolution. The system measuring the vertical particle flux used an ultrasonic 3d anemometer combined with a CPC at 23 m above ground. The CPC had a 10 nm cutoff size. The vertical particle flux was calculated using the eddy covariance method (Buzorius et al., 2000). When analyzing the May 8, 2013 case, we also used wind data from an ultrasonic 3d anemometer that was situated at 125 m above ground on top of the station's mast.

## 2.4 NPF event analysis

NPF event analysis, as described by Kulmala et al. (2012), was done for the flight measurement days (Kulmala et al., 2012). First the measurement days were classified into three different NPF event classes (NPF event days, undefined days and nonevent days) based on the DMPS data. NPF event days display a continuously and smoothly growing particle mode starting from the smallest detectable size. This indicates a regional NPF event. On undefined days sub-25 nm particles are only intermittently observed without apparent growth or a growing Aitken mode appears, possibly arising from a NPF episode elsewhere. On nonevent days no increase in sub-25 nm particle number concentration is observed.

Particle growth rate ($GR$) is the rate of change of particle diameter. We used the mode-fitting method to determine the particle $GR$s. The method involves fitting log-normal curves over the particle size distributions on the growing particle mode, defining the peaks as the mean particle diameters of the mode, and then using the change in the mean particle diameter with respect to time to calculate the $GR$.





The formation rate of diameter *d* particles is defined as the rate at which the freshly formed particles enter a certain size as a result of NPF. The formation rate $J_d$ was calculated using the below formula (Kulmala et al., 2012)

$$\frac{dN_d}{dt} = J_d - \frac{GR}{\Delta d} \times N_d - CoagS_d \times N_d$$

where $N_d$ is the number concentration of particles in the size range $\Delta d$, *GR* is the growth rate and $CoagS_d$ is the coagulation sink for the particles in the size range.

**2.8 Determination of BL height.**

The height of the BL was determined from the aircraft measurements by inspecting the vertical profiles
of relative humidity and potential temperature. The purpose was to determine if the roll-induced NPF events were observed inside the BL or above it. We determined the height of the BL to be approximately at the altitude where there was a minimum vertical gradient in relative humidity and a maximum vertical gradient in potential temperature (Seidel et al., 2010).

**2.9 Detection of roll vortices**

Inspecting satellite images for cloud streets was one way to deduce the presence of rolls (Etling and Brown, 1993). For this NASA's WorldView online tool was used. One limitation of this method was that clear sky rolls or rolls underneath a cloud cover could not be identified. Also the measurement flight time and the time of the satellite image were often separated by several hours and the meteorological conditions could change during that time.

The roll-axis can deviate from the mean BL flow direction (Miura, 1986) which causes the rolls to slowly move perpendicular to the mean BL flow direction leaving low-frequency periodic variation in the time series of the wind components when measured from a stationary point (Buzorius et al., 2001; Smedman, 1991). This provided one way to determine if roll circulation was taking place. The vertical
and parallel to roll-axis wind components would always be in phase opposition while the phases of the





perpendicular to roll-axis and parallel to roll-axis wind components would be separated by either 90 or -90 degrees depending on the direction of the roll movement (Brooks and Rogers, 1997; Vandemark et al., 2001). We used the mean horizontal wind component in place of the parallel to roll-axis wind component since they should not deviate that much from each other. The roll-induced variation in wind could be directly observed in the smoothed wind components measured on board the airplane by the turbulence probe.

Weather radars can detect clear air echoes, and insects are the most important source of echoes at these radio frequencies (Finnish C-band weather radars operate at 5.6 GHz) (Wainwright et al., 2017; Wilson et al., 1994). The lack of insects in the air limits the important period of the use of weather radar clear air echoes in Finland approximately from May to September. Organized convection causes the insects to congregate due to the lower BL convergence related to the updraft zones. The number density of insects in the updraft zone is probably further increased by the insects' tendency to resist upward motion to lower temperatures, adiabatic cooling of the rising air. As a result the weather radars show the maxima of upward motions as maxima of reflectivity (Wainwright et al., 2017).

In our case the Finnish Meteorological Institute weather radar in Ikaalinen (61°46'1.6"N, 23°4'47.6"E, 154 m above sea level) provided information on the existence and location of planetary BL rolls. The analysis of the radar data was based on the processed radar imagery. Most of the flight-tracks were in the range 50 to 70 km from the radar, and during the summer season insects are usually abundant enough to let the rolls be visible in the radar images over the area of airborne observations. The spatial resolution of the radar measurements is set by the antenna beamwidth and pulse duration. Ikaalinen radar resolution in range was 500 m, and the 1.0 degrees beam gets about 1 km wide over the target area. Some small rolls may get unresolvable, because of the radar resolution, but more probably the detection would have been limited already by the weakness of the circulation of these tiny rolls to get enough insects airborne high enough.



# 3 Results and discussion

Figure 2 shows a frequent observation in the measurement data: a momentary increase in the number concentration of freshly formed clusters and aerosol particles during daytime, coupled with a relatively large fluctuation in the vertical particle flux. Concurrent airplane measurements flown over the measurement station on that day showed that the location of increased aerosol particle number concentration was elongated along the mean wind direction, and that the maximum number concentrations occurred in two neighboring roll downdrafts (Figure 3). Increased number concentrations were not observed above the BL, no pollution sources were close-by and the sky was cloudless.

Wind measurements from the mast of the measurement station (Figure 4) showed that roll vortices were slowly moving perpendicular to the mean wind (this is due to a slight difference in the directions of the mean wind and the roll axis). This explains why particles were observed only momentarily at the field station, they were connected to specific rolls that drifted over the station. Overall, the observations on this day show that the roll circulation was locally inducing the formation of new aerosol particles.

We used two conditions to identify roll-induced NPF from the measurement data. Condition (i): a roughly 1-5 km wide region of increased sub-20 nm particle number concentration was observed on the flight track during consecutive overpasses when the airplane was flying perpendicular to the mean wind direction inside the BL. In other words this implies a long and narrow region of freshly formed particles inside the BL that is roughly aligned with the mean wind (see Figure 5 for examples). Condition (ii): in the ground-based measurements the number concentration of sub-20 nm particles momentarily (lasting between 0.5-2 hours) increased, and this increase was associated with opposite fluctuations in the vertical particle flux (see Figure 6 for examples). This would be due to the roll-induced NPF moving over the measurement station and it requires that the rolls are not aligned with the mean BL flow.

During the airborne measurement campaigns condition (ii) never occurred at the same time without condition (i) being also true, but condition (i) did occur without condition (ii). This is likely because



when the rolls were not aligned with the mean wind the roll-induced NPF could be observed from the airplane as well as from the station. Whereas if the rolls were aligned with the mean wind, then the roll-induced NPF could still be observed from the airplane but not from the measurement station.

The airborne measurement data was classifed with respect to NPF events and for the presence of roll vortices and roll-indcued NPF. The results are presented in Table 2. Roll-induced NPF was observed on 30% (6/20) of the regional NPF event days and on 22% (8/36) of the days classified as undefined (Figure 7A). According to radar and satellite observations the counter-rotating horizontal circulations were always present during the roll-induced NPF (Figure 7B) and this association was statistically significant ($p$=0.03). Roll vortices do not guarantee that roll-induced NPF occurs, since many other factors, such as a sufficient amount of sunlight and low enough sinks for low-volatile vapors and small clusters, are also important in determining whether atmospheric NPF may occur or not (Dada et al., 2017; Hamed et al., 2007).

The timescale that a roll-induced NPF moves over the measurement station is roughly an hour (see Table 3). This timescale is associated with mixing throughout the convective BL and it allows us to estimate the total effect of a roll on NPF. Using condition (ii), we identified some of the clearest cases of roll-induced NPF (29 days and 46 roll-induced NPF events) from only the ground-based measurements during 2006-2017 and summarized the results in Table 3. By looking at the change in particle diameter between subsequent roll-induced NPF events during the same day we found that the median $GR$ of the roll-induced NPF particles was 1.9 (inter quartile range (IQR) = 1.3-2.1) nm/h. On May 8, 2013 we could calculate the $GR$ from a single roll-induced NPF event by following it with the zeppelin aircraft (Figure 8). This is similar to the median $GR$ of 2.5 nm/h for 3-25 nm particles reported by Nieminen et al. (2014) for regional-scale NPF events observed at the station (Nieminen et al., 2014).

We aggregated all the roll-induced NPF observations in Table 3 into 1-hour-averaged bins (Figure 9) using the median GR and the geometric mean diameters of the particles. This was used in the calculation of particle formation rates. The resulting peak formation rate was 2.4 (IQR=1.6-3.1) cm$^{-3}$s$^{-1}$





for 3-nm particles and 0.4 (IQR = 0.2-0.6) cm$^{-3}$s$^{-1}$ for 10-nm particles. Nieminen et al. (2014) found that for regional-scale NPF events during springtime, the median formation rates of 3-nm and 10-nm particles were 1.0 cm$^{-3}$s$^{-1}$ and 0.52 cm$^{-3}$s$^{-1}$, respectively (Nieminen et al., 2014).

In addition, we estimated the fraction of area covered by the roll-induced NPF by dividing the time that the subsequent roll-induced NPF events observed during one day spent on top of the measurement station with the total time it took for the roll-induced NPF events to move over the site. We found that the fraction of area covered by the roll-induced NPF was 0.46 (IQR = 0.39-0.64). The roll systems are regionally roughly homogeneous (as demonstrated by cloud streets caused by the rolls in satellite images), so we can assume that the fraction of area covered by the roll-induced NPF events applies regionally and the phenomena is not limited to the close vicinity of the site.

We combined the median formation rates, the median area coverage and the statistics obtained from the aircraft campaigns to estimate how much, in terms of percentage increase, the roll-induced NPF enhances the production of new aerosol particles in Hyytiälä:

$$\text{NPF enhancement} = \frac{a \times n_{\text{roll-induced}} \times J_{\text{roll-induced}}}{n_{\text{regional}} \times J_{\text{regional}}} \times 100\%$$

where $a$ is the median area fraction of the roll-indcued NPF, $n$ is the number of roll-induced and regional NPF events observed and $J$ is the median formation rate of particles size $d$. The uncertainty was calculated by using the propagation of uncertainty. We estimate that compared with only regional NPF the roll-induced NPF enhances the production of new aerosol particles by 83±34% and 26±8% for 3-nm and 10-nm particles respectively. In addition to the enhancement of regional NPF, there were several days during which practically no NPF would have taken place without roll-induced NPF (see cases in Figure 5).

## 4 Conclusions

The processes that lead to roll-induced NPF are conceptualized in Figure 10. Roll vortices strongly
enhance mass transfer at the atmosphere-biosphere interface (Zilitinkevich et al., 2006). The narrow
updrafts collect and efficiently deliver low-volatile gases and clusters from the surface to the upper parts
of the BL where nucleation is more favorable due to lower temperatures and mixing over the inversion
(Buzorius et al., 2001; Easter and Peters, 1994; Nilsson et al., 2001; Nilsson and Kulmala, 1998). The
freshly-formed particles are then transported down by the downdrafts. We found that roll-induced NPF
can considerably enhance the production of new aerosol particles over a boreal forest and these particles
can grow to larger, potentially CCN, sizes, similar to particles produced by regional NPF. Roll-induced
NPF seems to occur in only some of the roll vortices, which is likely related to variability in the rolls.

NPF is a ubiquitous phenomenon in the global atmosphere (Kerminen et al., 2018; Kulmala et al.,
2004), likewise roll vortices are a common feature in the planetary BL around the world (Atkinson and
Wu Zhang, 1996; Etling and Brown, 1993; Young et al., 2002). Therefore, roll-induced NPF is expected
to take place in several other environments around the world as well. Hence, unstable stratification and
the formation of roll vortices needs to be taken into account in order to understand the overall role of
atmospheric NPF in particle number and CCN budgets.

**Author contributions.** JL, KL, RV, AM, SBM, HEM and JK carried out the airborne measurements.
ML processed and interpreted the weather radar data. MB, SZ and HJ helped with the analysis of
meteorological data. JL prepared the manuscript with contributions from all co-authors.

**Data availability.** The airborne and ground-based data from Hyytiälä used in this study were gathered
into a data set (doi:10.5281/zenodo.3688471). The processed weather radar images can be found in
http://www.atm.helsinki.fi/~mleskine/RADAR/pbl/rolls.html.

**Competing interests.** The authors declare that they have no conflict of interest.



**Acknowledgements.** This project has received funding from the ERC advanced grant No. 742206, the European Union's Horizon 2020 research and innovation program under grant agreement No. 654109, the Academy of Finland Center of Excellence project No. 272041 and the European Commission under the Framework Programme 7 (FP7-ENV-2010-265148). S. Zilitinkevich acknowledges support from the Academy of Finland projects ABBA No. 280700 (2014-2017) and ClimEco No. 314 798/799 (2018-2020); and Russian Science Foundation project No. 15-17-20009 (2015-2018). We appreciate the efforts that the Zeppelin NT pilots and ground crews made to this work. We acknowledge Dr. T. F. Mentel and Dr. F. Rohrer from Forschungszentrum Jülich, Germany. We thank Erkki Järvinen and the pilots at Airspark Oy for operating the research airplane. We acknowledge the use of imagery from the NASA Worldview application (https://worldview.earthdata.nasa.gov/) operated by the NASA/Goddard Space Flight Center Earth Science Data and Information System (ESDIS) project.





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

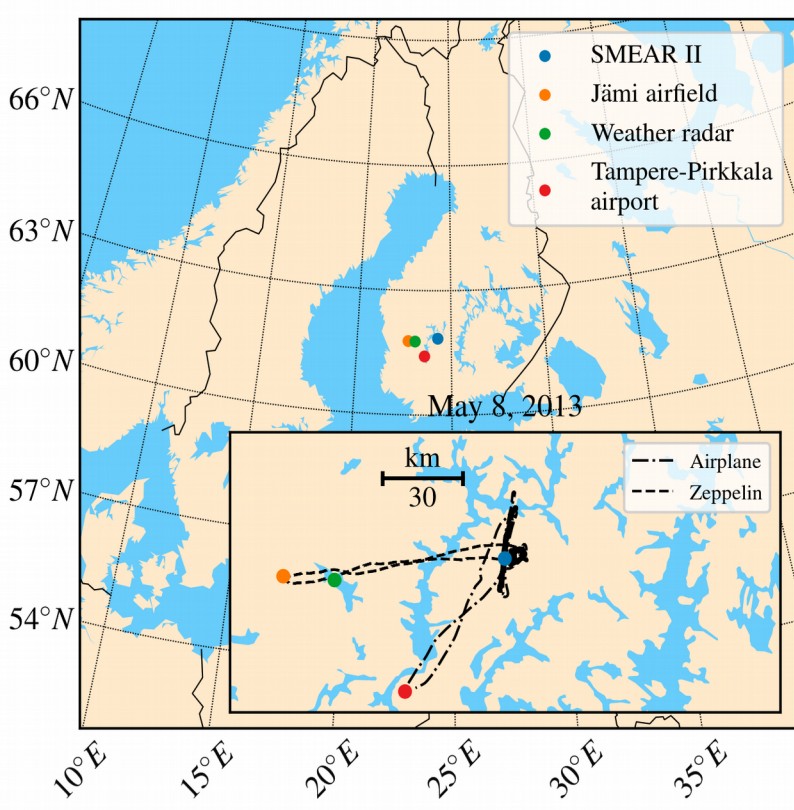

**Figure 1:** The locations of the Tampere-Pirkkala airport (ICAO: EFTP), Jämi airfield (ICAO: EFJM),
Ikaalinen weather radar and SMEAR II station marked on a map. As an example, the aircraft
measurement tracks on May 8, 2013 are included.





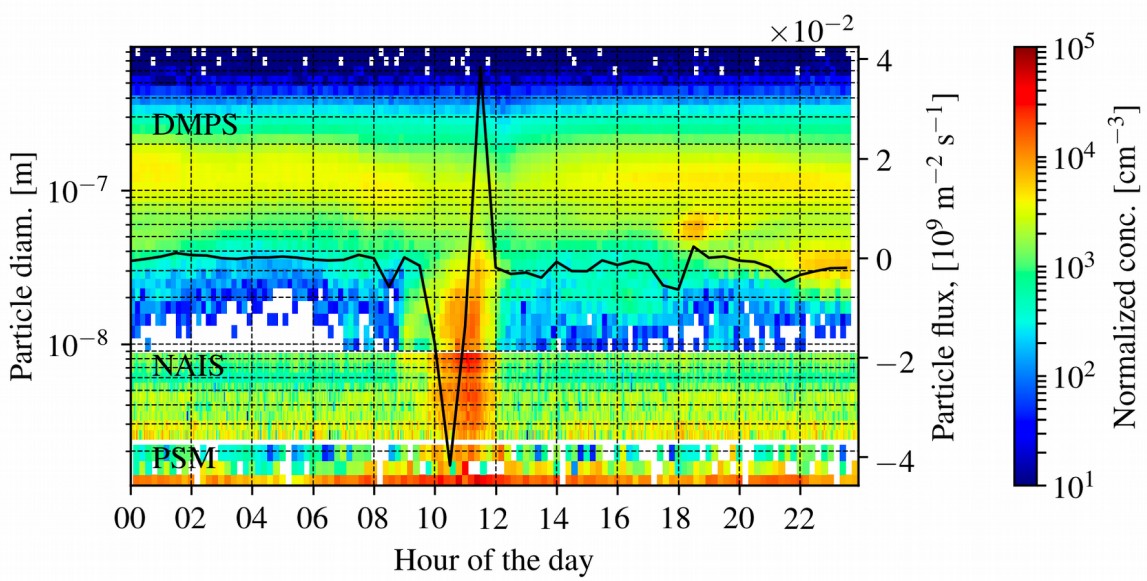

**Figure 2:** The particle number-size distribution in the range 1-1000 nm (composite of PSM, NAIS and DMPS data, see the Methods for instrument details) measured at the SMEAR II station on August 21, 2015. The black line is the vertical flux of >10 nm particles measured above the forest canopy (23 m height, negative sign means downward flux). Freshly formed clusters and aerosol particles were observed between 10:00 and 12:00. The particle number concentration for >1.5 nm particles increased from 2000 cm⁻³ in the background up to 18500 cm⁻³. Simultaneous airborne measurements over the SMEAR II station revealed that the number concentration maxima were linked to specific adjacent roll vortices (Figure 3).

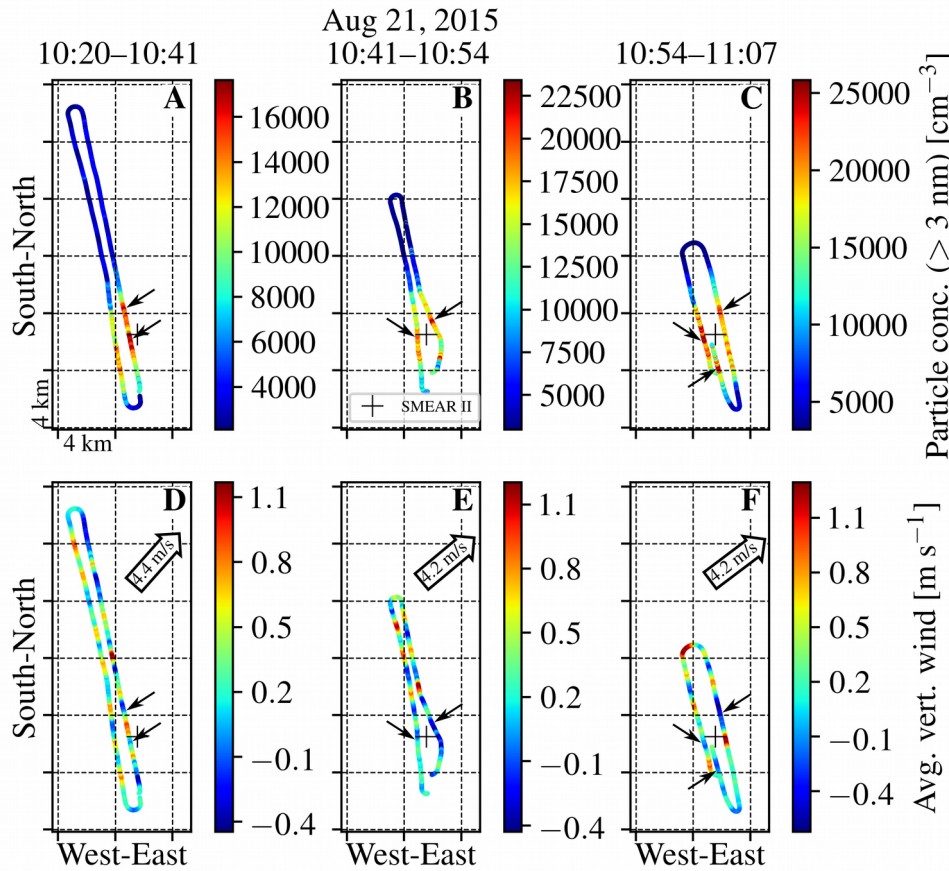

**Figure 3:** In panels A-C the sections of the measurement aiplane's flight track are colored by >3 nm particle number concentration. The grids have a 4-by-4 km spacing, the plus sign marks the position of the SMEAR II station and the time intervals for the flight track sections are displayed on top of the panels. In panels D-F the same flight tracks are colored by vertical wind speed smoothed using 30-sec moving average. The positive sign refers to updraft and the negative sign to downdraft. The large arrows show the mean wind speed and direction measured on board the airplane. The flight tracks were flown inside the convective BL between 120 m and 620 m above ground. The roll-induced NPF was observed in the southern part of the flight track directly on top of the SMEAR II station. The vertical wind speed measurements on board the airplane revealed the presence of rolls as regularly alternating up- and downdrafts that were aligned with the mean wind, over the flight path. The small arrows show that the maxima in the particle number concentration were located in the roll downdrafts.



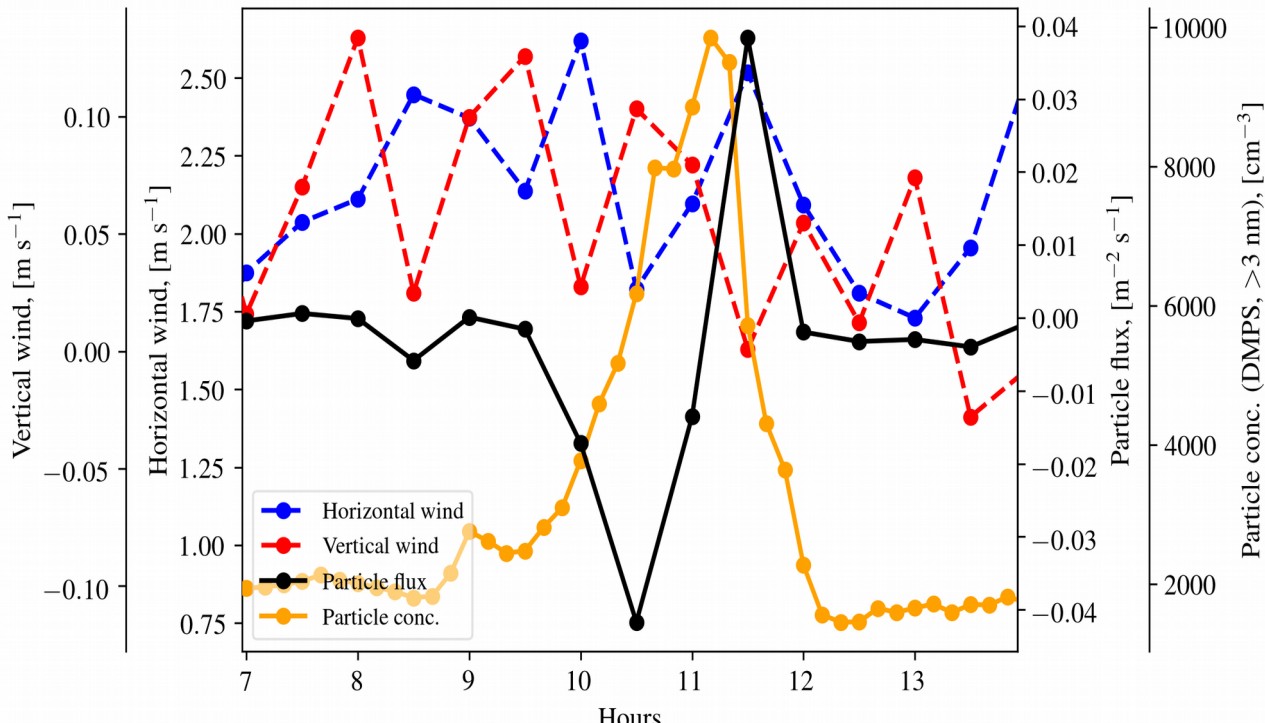

**Figure 4:** Time series of the vertical wind, horizontal wind, vertical flux of >10 nm particles (averaged to 30 min, measured at 23 m above ground or 1-2 m above canopy) and number concentration of >3 nm particles inside the canopy on Aug 21, 2015. The periodic anti-correlation between the wind components is a clear indication of roll vortices drifting over the measurement location perpendicular to the mean wind direction. This is due to a slight difference between the direction of the mean wind and the roll axis. During a sunny August day with moderate wind, turbulence dominates vertical transport close to the canopy, so the variations in particle number concentration and vertical particle flux close to the canopy are decoupled from the roll circulation. When the roll-induced NPF first moves over the field site the number concentration above the turbulent layer increases and the particles start to mix downwards. Inside the turbulent layer the particle flux becomes negative and the number concentration starts to increase. As more and more particles are mixed downwards, the number concentration increases inside the turbulent layer while the particle flux becomes less negative. As the roll-induced NPF moves away, the vertical particle flux can become positive if the number concentration below the flux measurement is higher than above.


**Figure 5:** Panels A-D show the research airplane's flight tracks colored by particle number concentration in the 3-20 nm diameter range on four different measurement flights. The higher particle number concentrations are displayed on top in order to make the roll-induced NPF more clearly visible. The locations of roll-induced NPF were observed as narrow areas of increased particle number concentration perpendicular to the mean wind direction that persisted over multiple successive overpasses. They were not seen when the airplane was flying above the convective BL. Above the BL the particle number concentration was about an order of magnitude lower. The arrows show the mean wind direction and speed from the SMEAR II mast.



**Figure 6:** The panels A-H show 3-1000 nm particle number-size distribution measured at the SMEAR II station by the DMPS during some of the days when there was roll-induced NPF. In addition the black line shows the vertical flux of >10 nm particles measured at 23 m height. The roll-induced NPF was marked by momentary increase in sub-20 nm particles, coupled with a relatively large fluctuation in vertical particle flux compared to background. If not enough of the particles were above 10 nm, then no clear fluctuation in the particle flux can be seen.





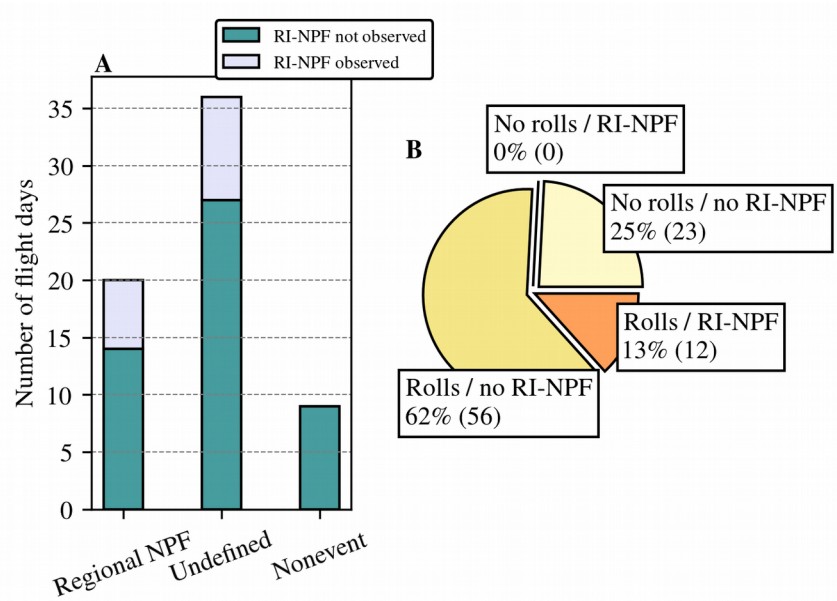

**Figure 7:** Panel A shows that roll-induced NPF (RI-NPF) was observed on 23% of the flight measurement days, 30% of the NPF event days and 22% of the days classified as undefined. Panel B shows the association between roll-induced NPF and roll observations from radar and satellite images. The data in panel A consists of flight days while the data in panel B consists of individual flights (note that there could be two flights per day, see Table 1). Rolls were always present during roll-induced NPF and according to Fisher's exact test this association was significant ($p$=0.03). The fact that rolls did not induce NPF every time means that other necessary factors for atmospheric NPF (e.g. photochemistry, sinks) were likely not satisfied.





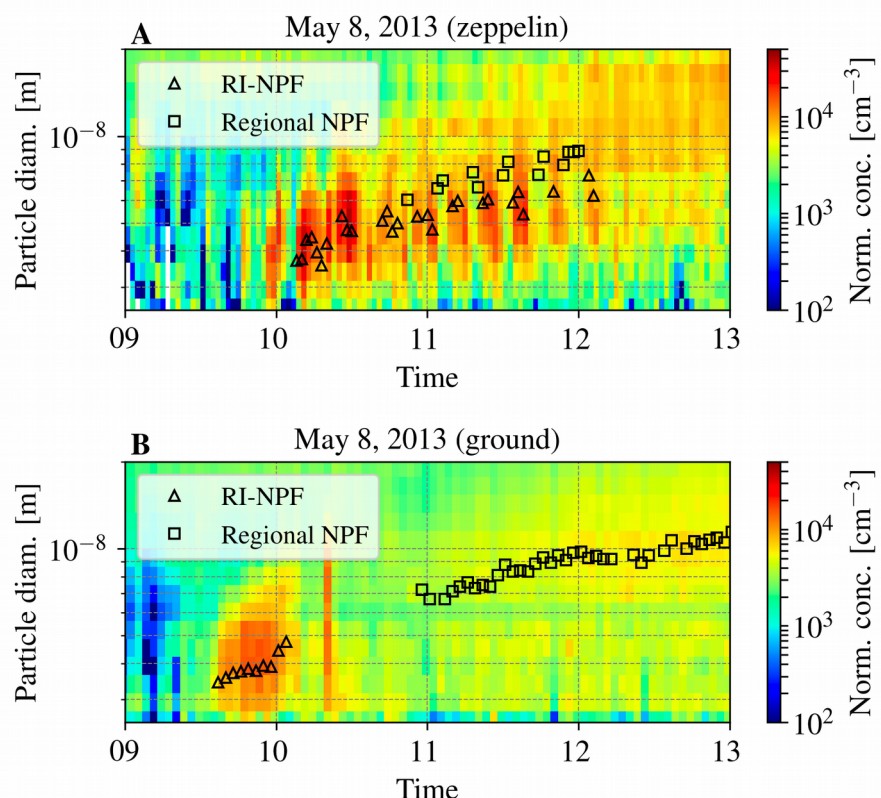

**Figure 8:** The particle number-size distribution (positive polarity) between 2.5-20 nm measured by the
NAIS (A) on board the zeppelin and (B) at the field station on May 8 2013. Between 10-12 the zeppelin
consecutively flew through the roll-induced NPF (RI-NPF) event, leaving concentrated "stripes" on the
particle number-size distribution. Between 9:30-10:00 the roll-induced NPF event moved over the field
station. The black triangles and squares show the fitted mean mode diameters to the roll-induced and
regional NPF event particles, respectively. Figure 5B shows simultaneous observations from the
airplane, however there were no wind measurements on board. Weather radar observations showed that
rolls were present over the measurement site and power spectra of the wind components from the
station's mast showed that the rolls were moving over the site.

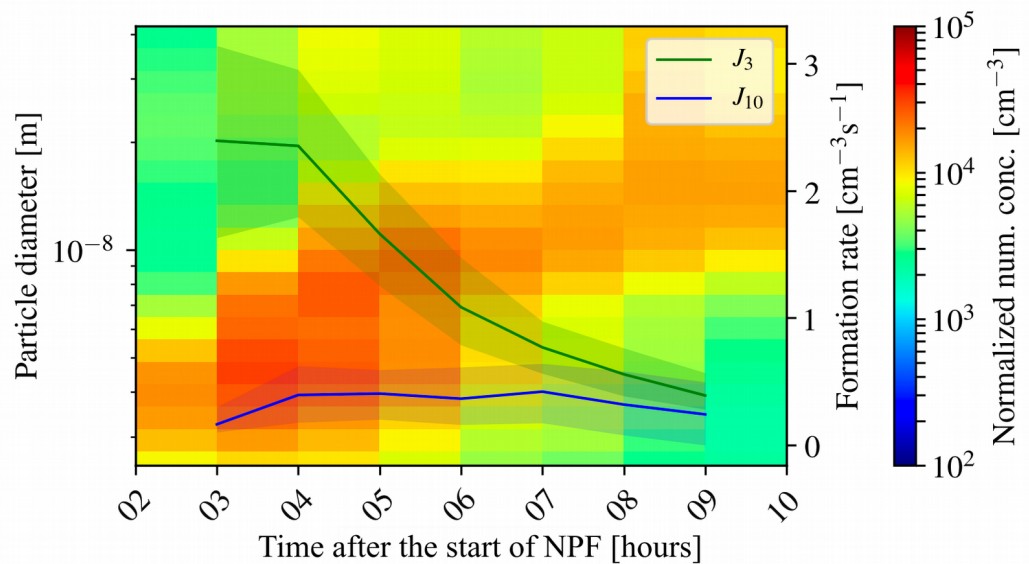

**Figure 9:** The above particle number-size distribution was constructed using the SMEAR II station's NAIS data by taking the roll-induced NPF observations presented in Table 3 (29 days and 46 different roll-induced NPF events) and distributing them along the time axis according to their geometric mean diameter while assuming growth rate of 1.9 nm/h, which was calculated from days that showed multiple subsequent roll-induced NPF events (13 days). The resulting number-size distribution was averaged to 1-hour bins and the variance in each bin was noted. We then used random sampling (1000 samples), also varying the GR, to estimate 25th, 50th and 75th percentile values for the formation rates of 3- and 10-nm-sized particles.


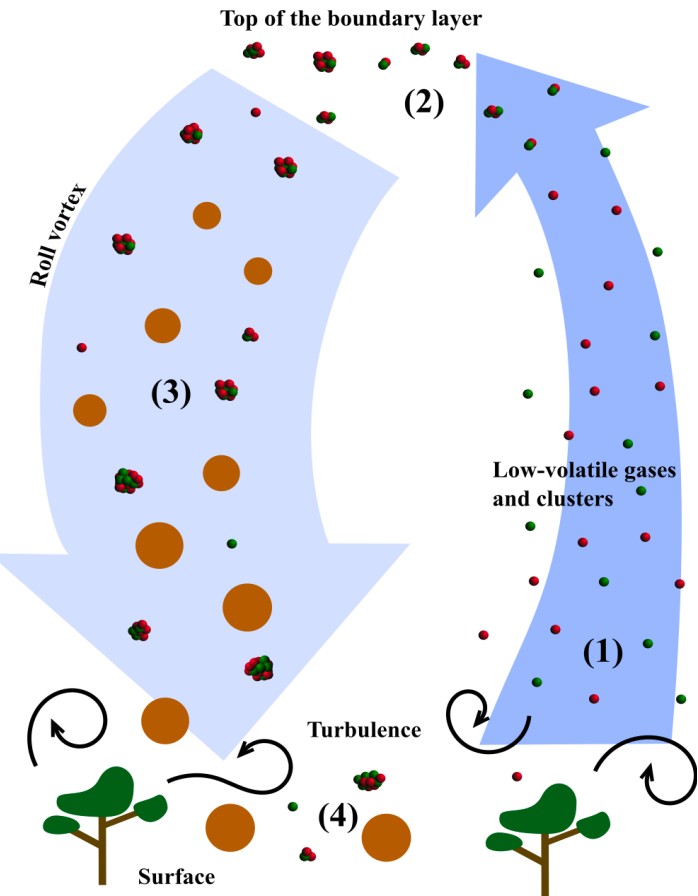

**Figure 10:** A schematic illustration of roll-induced NPF. The arrows depict the updraft and the downdraft zones of a single roll vortex, viewed along the roll. (1) in the boreal forest the vegetation is an important source of volatile organic compounds that can be oxidized into low-volatile organic vapors (Ehn et al., 2014). Due to higher wind speeds the shear-generation of turbulence close to the vegetation is stronger in rolls than in cellular type convection (Zilitinkevich et al., 2006). Therefore, roll updrafts are particularly efficient at transporting vapors and molecular clusters from the surface to the top of the BL. (2) on top of the BL decreased temperature, turbulence and mixing over the inversion layer can lead to a supersaturation of the vapors and activation of the clusters, leading to subsequent NPF (Easter and Peters, 1994; Nilsson and Kulmala, 1998). (3) the newly-formed particles grow in size in the weaker and wider downdraft and end up close to the surface where (4) they may be deposited on surfaces or continue growing while being transported in the air.





**Table 1:** Summary of airborne measurement campaigns from which data was utilized in this study. Explanations: PNSD = particle number-size distribution, INSD = ion number-size distribution, PNC = particle number concentration.

| Time Place | Number of flight days | Measurement platform(s) | Instruments on board the aircrafts that were used in this study |
|---|---|---|---|
| May-Jun 2013 Hyytiälä, Finland | 26 | Zeppelin NT Cessna 172 | **Zeppelin NT**<br>▪ NAIS: 2-42 nm PNSD and 0.8-42 nm positive and negative INSD<br>▪ Meteorological sensors: static pressure, temperature and relative humidity<br>**Cessna 172**<br>▪ TSI 3776 CPC: >3 nm PNC<br>1. SMPS: 10-400 nm PNSD<br>2. Li-Cor Li-840: $CO_2$ and $H_2O$ vapor concentration<br>3. Meteorological sensors: static pressure, temperature and relative humidity<br>**Cessna 172 (Aug 2015, last half of the campaign)**<br>● AIMMS-20: 3d wind vector |
| Mar-Apr 2014 Hyytiälä, Finland | 12 | Cessna 172 | |
| May-Jun 2014 Hyytiälä, Finland | 5 | Cessna 172 | |
| Aug-Sept 2014, Hyytiälä, Finland | 6 | Cessna 172 | |
| May-Jun 2015 Hyytiälä, Finland | 7 | Cessna 172 | |
| Aug 2015 Hyytiälä, Finland | 9 | Cessna 172 | |





**Table 2.** A summary of the flight campaign observations. Explanations: AM=morning flight, PM=afternoon flight, N=not observed, I=roll-induced NPF observed from the airplane, II=roll-induced NPF observed from the field station, R=roll vortices present over the measurement area, C=clear air (presence of rolls inconclusive), RE=regional NPF event observed, UD=undefined day, NE=nonevent day.

| Flight | Roll-induced NPF | Rolls | NPF event class |
|---|---|---|---|
| 20130506 AM | N | R | RE |
| 20130506 PM | N | R | |
| 20130507 AM | N | R | UD |
| 20130507 PM | N | R | |
| 20130508 AM | I/II | R | RE |
| 20130508 PM | N | R | |
| 20130511 AM | I/II | R | UD |
| 20130511 PM | N | R | |
| 20130514 AM* | II | R | UD |
| 20130515 AM | N | R | RE |
| 20130516 AM | I | R | RE |
| 20130516 PM | I | R | |
| 20130517 AM | N | R | NE |
| 20130518 AM | N | N | UD |
| 20130520 AM | N | R | UD |
| 20130521 AM | N | R | UD |
| 20130522 AM | I | R | UD |
| 20130522 PM | N | N | |





| 20130523 AM | I/II | R | UD |
|---|---|---|---|
| 20130523 PM | N | N | |
| 20130525 AM | I | R | UD |
| 20130526 AM | N | R | UD |
| 20130526 PM | I | R | |
| 20130528 AM | N | R | NE |
| 20130529 AM | N | R | UD |
| 20130602 AM | N | R | UD |
| 20130602 PM | N | R | |
| 20130603 AM | N | R | UD |
| 20130603 PM | N | N | |
| 20130605 PM | N | N | UD |
| 20130606 AM | N | R | UD |
| 20130606 PM | N | N | |
| 20130608 AM | N | N | UD |
| 20130608 PM | N | N | |
| 20130609 AM | N | R | UD |
| 20130610 AM | N | R | UD |
| 20130610 PM | N | N | |
| 20130613 AM | N | R | UD |
| 20130613 PM | N | R | |
| 20130615 AM | N | R | RE |
| 20140325 AM | N | C | RE |





| 20140325 PM | N | C | |
| 20140326 AM | N | C | RE |
| 20140326 PM | N | C | |
| 20140327 AM | N | C | RE |
| 20140327 PM | I/II | C | |
| 20140328 AM | I | C | RE |
| 20140328 PM | N | C | |
| 20140331 AM | N | C | RE |
| 20140331 PM | N | C | |
| 20140401 AM | N | N | RE |
| 20140401 PM | N | N | |
| 20140402 AM | I/II | R | UD |
| 20140402 PM | N | R | |
| 20140403 AM | N | N | RE |
| 20140403 PM | N | N | |
| 20140407 AM | N | N | UD |
| 20140408 AM | I | C | RE |
| 20140408 PM | N | C | |
| 20140409 AM | I | C | RE |
| 20140409 PM | I | C | |
| 20140410 AM | N | C | RE |
| 20140410 PM | N | C | |
| 20140522 AM | N | R | UD |



| | | | |
|---|---|---|---|
| 20140523 AM | N | N | UD |
| 20140523 PM | N | R | |
| 20140602 AM | N | R | RE |
| 20140602 PM | N | R | |
| 20140604 AM | N | N | UD |
| 20140605 AM | N | N | UD |
| 20140605 PM | N | R | |
| 20140822 AM | N | C | NE |
| 20140822 PM | N | N | |
| 20140827 AM | N | R | NE |
| 20140909 PM | N | R | NE |
| 20140910 AM | N | N | NE |
| 20140911 AM | N | R | NE |
| 20140915 AM | N | R | RE |
| 20140915 PM | N | R | |
| 20150527 PM | N | R | NE |
| 20150528 AM | N | R | UD |
| 20150528 PM | N | R | |
| 20150604 PM | N | C | RE |
| 20150604 AM | N | N | |
| 20150605 PM | N | C | RE |
| 20150605 AM | N | R | |
| 20150608 PM | N | R | UD |



| | | | |
|---|---|---|---|
| 20150608 AM | N | R | |
| 20150609 PM | N | R | UD |
| 20150609 AM | N | R | |
| 20150610 PM | N | R | UD |
| 20150610 AM | N | R | |
| 20150813 AM | N | R | RE |
| 20150813 PM | N | R | |
| 20150814 AM | N | R | UD |
| 20150814 PM | N | R | |
| 20150817 AM | N | R | UD |
| 20150817 PM | I | R | |
| 20150818 AM | N | N | UD |
| 20150818 PM | N | N | |
| 20150819 AM | N | N | UD |
| 20150820 AM | N | R | UD |
| 20150821 AM | I/II | R | UD |
| 20150821 PM | N | R | |
| 20150824 AM | N | N | UD |
| 20150824 PM | N | R | |
| 20150825 AM | N | R | NE |
| 20150825 PM | N | R | |

*On May 14, 2013 a roll-induced NPF event was observed from the field station after the flight.



**Table 3:** Summary of the ground-based roll-induced NPF observations. BT = begin time of roll-induced NPF observation, ET = end time of roll-induced NPF observation, $D_p$ = geometric mean particle diameter of roll-induced NPF event, Coverage = the time that subsequent roll-induced NPF events spent on top of the measurement station divided the total time it took for the subsequent roll-induced NPF events to move over the measurement site.

| Date | BT [hours] | ET [hours] | $D_p$ [nm] | GR [nm/h] | Coverage |
|---|---|---|---|---|---|
| 20060921 | 11:50 | 12:57 | 13 | | |
| 20070416 | 12:30<br>16:49<br>19:49 | 15:11<br>18:25<br>21:10 | 7<br>15<br>20 | 1.9 | 0.65 |
| 20070610 | 15:10<br>19.21 | 15:50<br>20:00 | 12<br>30 | 4.3 | 0.27 |
| 20090510 | 13:59 | 15:23 | 12 | | |
| 20100312 | 10:57 | 11:46 | 4 | | |
| 20100418 | 15:10 | 16:14 | 9 | | |
| 20100419 | 10:36 | 11:55 | 7 | | |
| 20110602 | 12:15<br>16:58 | 13:29<br>17:55 | 15<br>21 | 1.3 | 0.39 |
| 20110912 | 11:29 | 12:54 | 7 | | |
| 20110930 | 16:47 | 18:20 | 9 | | |
| 20120328 | 10:24<br>13:57<br>17:05 | 11:03<br>14:55<br>18:04 | 4<br>10<br>12 | 1.2 | 0.34 |
| 20120331 | 12:28 | 14:24 | 7 | | |
| 20120405 | 11:08<br>14:29 | 12:02<br>15:53 | 9<br>12 | 1.0 | 0.48 |
| 20120409 | 10:42 | 13:13 | 7 | | |
| 20120430 | 13:09 | 14:47 | 9 | 2.6 | 0.78 |





| | | | | | |
|---|---|---|---|---|---|
| | 15:37 | 16:56 | 15 | | |
| 20130308 | 15:41 | 16:24 | 6 | | |
| 20130328 | 12:04 16:03 | 13:03 17:35 | 6 10 | 0.8 | 0.46 |
| 20130508* | 09:36 | 10:09 | 4 | 1.8 | |
| 20130511 | 13:03 16:12 | 14:23 18:00 | 12 19 | 1.8 | 0.64 |
| 20130514 | 13:53 | 15:07 | 13 | | |
| 20130523 | 08:13 10:29 16:55 20:00 | 8:57 11:00 17:26 20:34 | 9 12 18 20 | 2.1 | 0.19 |
| 20140327 | 13:40 16:27 | 15:21 18.28 | 5 15 | 3.6 | 0.77 |
| 20140402 | 09:34 12:21 | 10:36 13:17 | 5 8 | 2.0 | 0.53 |
| 20150821 | 09:44 | 11:57 | 10 | | |
| 20150913 | 09:59 14:48 | 10:59 16:43 | 8 18 | 1.9 | 0.43 |
| 20160415 | 13:07 | 15:15 | 6 | | |
| 20170324 | 15:15 | 17:39 | 7 | | |
| 20170424 | 13:39 17:15 | 14:44 18:05 | 4 11 | 2.0 | 0.43 |
| 20170604 | 11:23 | 12:50 | 7 | | |

*For May 8, 2013 the GR was determined from the zeppelin data when the zeppelin was consecutively flying through the roll-induced NPF event (Fig. E5).