# Peer review of "Roll vortices induce new particle formation bursts in the planetary boundary layer"

_Atmospheric Chemistry and Physics, 2019_

## Referee Comment (RC1) · Anonymous Referee #1 · 13 May 2020

This paper describes the influence of BL dynamics induced by roll vortices on new particle formation. To do so, the authors used a large data set issued from ground-based and aboard instruments devoted to Ultra-Fine Particles (UFP) particle size distribution measurements (Particle magnifier, NAIS, DMPS) and wind measurements. They found that roll induced events are numerous over the boreal forest and need to be well simulated to better understand nucleation processes within the atmosphere.

Major comments

1. Measurement average : All measurements are performed at different frequency and the authors choose to average all the data over different periods ( 4 min, 12 min, 30min). They did not justify why they choose these periods. Why not using the same periods for all instruments ? The DMPS SD are not averaged what was the frequency

for this instrument ?

2. Measurement location : Could you please justify that inlet height differences are unnoticeable on aerosol measurements ? Especially, when you used the synergy between anemometer at 125m above the ground with a CPC at 23m.

3. In general the figure labels are really long because you explained most of the time the way you used to process the data. I found it odd, especially because you are limited in word numbers. For example, I have many questions about Figure 9. From what I understood, figure 9 shows size distribution and formation rate calculations based on observations of geometric mean diameter.

- First of all, geometric mean diameter given in table 3 is observed at the end of the event, an hour after the beginning ? This is not clearly stated when the Dp values refer to.

- Then you use a constant GR of 1.9nm/h. Why ? You have measured the GR for each case. Then why using this value corresponding to GR from days that showed multiple subsequent roll-induced NPF events ? According to Table 3, the GR ranges from 0.8 to 4.3 nm/h. The use of GR value 2 times lower or larger might causes a lot of difference in the diameter growth and the formation rate.

- Moreover, I don't understand the last sentence : Âń We then used random sampling (1000 samples), also varying the GR, to estimate 25th, 50th and 75th percentile values for the formation rates of 3- and 10-nm-sized particles Âż . From this sentence, one can understand that the GR is not fixed anymore. What are the values used then ??? Also, you used 1000 random samples from what you calculated. Do you have 1000 samples from what you calculated ? You have 3 (GR variations ?) * One SD/hour *nb of events (46) or did I miss something ?

- How do you control this random factor ? Could the 1000 samples belongs to one or 2 specific events ? If the GR is two times larger, what will be the error on the formation

rate.

- And so you did all that to get formation rates that you measured directly ???

4. Fraction of area : So you use a ratio of two periods and that give you a fraction area covered by the roll-induced NPF. Could you please explain the idea behind it ? I guess that this is related to the wind speed of the air mass over the site vs over the region. So, assuming both wind speeds are similar this is just a ratio of the horizontal extend of the NPF event when passing over the site and the horizontal extend of the NPF observed by the airborne instruments.

What is the time shift between the aboard and grounded measurements ?

Is the wind speed really constant during the whole period ?

Minor remarks

L161 – 171 : You could probably use figure 10 to ease the understanding. It would be useful !

L176 : "Organized convection causes the insects to congregate due to the lower BL convergence related to the updraft zones. The number density of insects in the updraft zone is probably further increased by the insects' tendency to resist upward motion to lower temperatures, adiabatic cooling of the rising air."

Please rephrase these two sentences. I think there are many ideas in there but need to be further explained. Personally, I don't know anything about insects and this is hard to link it to the dynamics you seemed to describe.

L 225 : induced not induced

Figure 8 : These two figures are pretty interesting but I think that you need the reader to understand what you show. So here there are apparently 2 event types : One regional and one induced by roll vortices. Looking at Figure 5b, I see several zones associated with high N3-20. One in the 4 first km north to SMEAR II mast and the second one is

further north (12km). According to wind speed direction the one located further north did not cross the site measurement. So my question is how could you separate the Roll vortices induced NPF from the regional one given the fact that both are located in the same zone ? If you used only the mean geometrical diameter, could you please justify why this is relevant ???

---

## Author Comment (AC1) · 30 May 2020

We thank the referee for the constructive comments on our manuscript, please find our responses below.

Major comments

1. Measurement average : All measurements are performed at different frequency and the authors choose to average all the data over different periods ( 4 min, 12 min, 30min). They did not justify why they choose these periods. Why not using the same periods for all instruments ? The DMPS SD are not averaged what was the frequency for this instrument?

Answer: The time resolutions used were those of the processed data, we did not reaverage the processed data afterwards. We changed the text to better reflect this. The DMPS processed data had time resolution of 10 min, we also added this information to the text.

"The NAIS measured the particle number-size distribution in the mobility diameter range 2-42 nm and ion number-size distribution in the mobility diameter range 0.8-42 nm at 4 min time resolution. We used data from the positive polarity of the instrument."

"The time resolutions of the DMPS and the NAIS were 10 min and 4 min respectively, from the NAIS we again used particle data from the positive polarity. The PSM measured particle number-size distribution between 1-2 nm and the time resolution was 12 min."

2. Measurement location : Could you please justify that inlet height differences are unnoticeable on aerosol measurements ? Especially, when you used the synergy between anemometer at 125m above the ground with a CPC at 23m.

Answer: The vertical particle flux was calculated from a CPC and a 3d anemometer positioned at 23 m above ground. The anemometer at 125 m above ground was only used in wind data analysis. We removed the mention of the 125 m anemometer from the text since the results of this analysis are not shown in the text.

3. In general the figure labels are really long because you explained most of the time the way you used to process the data. I found it odd, especially because you are limited in word numbers. For example, I have many questions about Figure 9. From what I understood, figure 9 shows size distribution and formation rate calculations based on observations of geometric mean diameter.

Answer: We made the captions in Figures 2-9 shorter by moving some of the interpretation of the figure to the main text instead of keeping it in the caption. Figure 8 caption we kept the same since it describes a supporting case study and explaining it in detail in the main text would break the flow where it is mentioned as part of the GR

estimation.

- First of all, geometric mean diameter given in table 3 is observed at the end of the event, an hour after the beginning ? This is not clearly stated when the Dp values refer to.

Answer: At the measurement station the roll vortex induced new particle formation (RI-NPF) is observed as an intermittent, concentrated mode of sub-20 nm particles, with sudden beginning and end. One might describe it as a particle stripe in the sub-20 nm sizes. To obtain the geometric mean diameter for each RI-NPF reported in Table 3 we fitted a log-normal curve to the particle number-size distributions present in the RI-NPF and chose the peak value at the beginning of the RI-NPF observation as the geometric mean diameter that we report in Table 3.

In Table 3 caption we added that the geometric mean diameter is reported at the beginning of the RI-NPF:

"Dp = geometric mean particle diameter of roll-induced NPF event, determined at the beginning of the roll induced NPF observation"

- Then you use a constant GR of 1.9nm/h. Why ? You have measured the GR for each case. Then why using this value corresponding to GR from days that showed multiple subsequent roll-induced NPF events ? According to Table 3, the GR ranges from 0.8 to 4.3 nm/h. The use of GR value 2 times lower or larger might causes a lot of difference in the diameter growth and the formation rate.

- Moreover, I don't understand the last sentence : Âń We then used random sampling (1000 samples), also varying the GR, to estimate 25th, 50th and 75th percentile values for the formation rates of 3- and 10-nm-sized particles ÂÅij . From this sentence, one can understand that the GR is not fixed anymore. What are the values used then ??? Also, you used 1000 random samples from what you calculated. Do you have 1000 samples from what you calculated ? You have 3 (GR variations ?) * One SD/hour *nb

of events (46) or did I miss something ?

- How do you control this random factor ? Could the 1000 samples belongs to one or 2 specific events ? If the GR is two times larger, what will be the error on the formation rate.

- And so you did all that to get formation rates that you measured directly ???

Answer: In order to clearly observe particle growth we had to see more than one RI-NPF event go over the station. We estimated the GR from the change in the geometric mean diameters in the subsequent RI-NPF events. This happened on 13/29 days in Table 3. In addition on 8 May 2013 the zeppelin flew through the same RI-NPF multiple times throughout the day for several hours, which also allowed us to observe the particle growth and calculate GR.

We changed the text to read:

"Multiple roll-induced NPF events during a single day were observed on 13/29 days. In these cases by looking at the change in particle diameter between subsequent roll-induced NPF events we were able to estimate the GR. In addition, on May 8, 2013 we could calculate the GR from a single roll-induced NPF event by following it with the zeppelin aircraft (Figure 8)."

From the 14 GR values we calculated the median, 25th and 75th percentile GRs. These represent the average GR for the RI-NPF particles in Table 3. We assumed that the underlying distribution of the GRs is a normal distribution with mean equal to the median GR and the standard deviation equal to the inter quartile range (IQR) of the GRs. From this normal distribution we then randomly sampled a GR.

Assuming that in Table 3 the particles in each RI-NPF case were formed at time t = 0 hours and that the GR remained constant, we estimated the time dt since the RI-NPF particles in Table 3 were formed using dt = Dp/GR, where Dp is the geometric mean diameter of the particles at the beginning of the RI-NPF observation.

This way we were able to put all the RI-NPF observations in Table 3 on a common time axis where the time is the time since particle formation. We then divided this time axis into 1-hour bins and in each bin calculated the median, 25th and 75th percentile particle number-size distribution. Again we assumed that the particle number-size distributions in each bin were normally distributed with mean equal to the median and standard deviation equal to the IQR. Then we randomly sampled a distribution from each bin and used the randomly sampled values to calculate the formation rate time series for 3 nm and 10 nm particles.

The particle size distribution displayed in Figure 9 consists of medians in each bin and the median GR was used to calculate the time since start of NPF.

We repeated the above random sampling 1000 times in order to obtain 1000 formation rate time series. From these formation rates we calculated the median, 25th and 75th percentile values. These are then our estimates for the average J3 and J10 plus their uncertainties, which are displayed in Figure 9.

In principle we could calculate formation rate for some individual cases. This means the case needs to have at least two subsequent RI-NPF events during the same day in order to estimate the GR, and also there needs to be particles for long enough time in the interesting size-range. Being this specific discards most of the data and we are left with just a couple of case studies. Instead we wanted to use a method that uses all the available observations. This allows us to get a formation rate that better represents the average and allows us to estimate the uncertainty.

We added a more explicit description to the text regarding the above procedure.

"We aggregated all the roll-induced NPF observations in Table 3 into 1-hour-averaged bins using the median GR and the geometric mean diameters of the particles, assuming that the particles were formed at t=0 hours (Figure 9).

Then we calculated the formation rates and their uncertainties. We assumed that the

roll-induced NPF GRs were normally distributed with mean equal to the median GR and standard deviation given by the magnitude of the IQR. Given the sampled GR we distributed the roll-induced NPF observations into 1-hour bins. For each 1-hour bin we assumed that the number-size distributions again followed a normal distribution with mean equal to the median and standard deviation given by the IQR. We randomly sampled a number-size distribution from each bin and calculated the formation rates based on that. We repeated this procedure 1000 times in order to estimate the J3 and J10 and their uncertainties shown in Figure 9."

4. Fraction of area : So you use a ratio of two periods and that give you a fraction area covered by the roll-induced NPF. Could you please explain the idea behind it ? I guess that this is related to the wind speed of the air mass over the site vs over the region. So, assuming both wind speeds are similar this is just a ratio of the horizontal extend of the NPF event when passing over the site and the horizontal extend of the NPF observed by the airborne instruments. What is the time shift between the aboard and grounded measurements ? Is the wind speed really constant during the whole period ?

Answer: First we assume that the RI-NPF extends a long distance along the length of the rolls, which is supported by the aircraft data. So then to estimate the area fraction we want to know what the spacing of RI-NPF is perpendicular to the rolls.

We needed more than one RI-NPF observation at the station during the same day in order to estimate this. Figure 1 shows how the rolls and by extension the RI-NPF move over the station if there is a difference in the direction of the roll axis and the mean wind direction.

If the wind conditions stay the same during the period when the multiple RI-NPF events move over the measurement station, then we can assume that the rolls move over the site at a steady pace. This means that the spatial extent across the RI-NPF events is directly proportional to the time interval we observe the RI-NPF events at the field station. This means that the time that subsequent RI-NPF events spent on top of the

measurement station divided by the total time it took for these RI-NPF events to move over the station is equal to the fraction of area covered by the RI-NPF events.

We can check the wind measurements from the mast at 33.6 m height above ground and see how constant they are (the 125 m measurement was not available for the whole time). For this we prepared Figure 2. Of course this does not tell us how the wind behaves in the rest of the boundary layer.

On most of the days the wind conditions do not fluctuate significantly during the multiple RI-NPF observations. On 2007-06-10 and 2017-04-24 the wind direction changes more than 100 degrees, and this could introduce some uncertainty, but would not have much effect on the final result.

The above analysis only requires ground-based observations. Since the flights covered a relatively small area we found them to be inadequate at estimating the spatial extent of RI-NPF in the direction perpendicular to rolls. One might argue that along the flight tracks in Figure 5 the concentrated particle areas took roughly half of the area on the track, which is in line with our findings from the above analysis.

We changed the text to better explain the method:

"In addition, we estimated the fraction of area covered by the roll-induced NPF. We assumed that the roll-induced NPF events extend much longer along the rolls, which is supported by the aircraft data. This means that for the area fraction we need to estimate what the spacing of the roll-induced NPF events is perpendicular to the direction of the rolls.

If the wind conditions stay the same during the period when the multiple RI-NPF events move over the station, then we can assume that the rolls move over the station at a steady pace. This means that dividing the time that subsequent roll-induced NPF events observed during the same day spent on top of the measurement station by the total time it took for the roll-induced NPF events to move over the site can be used as

an area fraction estimate. According to measurements from the mast, on average the wind conditions during the observations did not change significantly."

Minor remarks L161 – 171 : You could probably use figure 10 to ease the understanding. It would be useful !

Answer: We prepared Figure 1 to illustrate how rolls move over the measurement station perpendicular to the mean wind direction and added it to the text.

L176 : "Organized convection causes the insects to congregate due to the lower BL convergence related to the updraft zones. The number density of insects in the updraft zone is probably further increased by the insects' tendency to resist upward motion to lower temperatures, adiabatic cooling of the rising air." Please rephrase these two sentences. I think there are many ideas in there but need to be further explained. Personally, I don't know anything about insects and this is hard to link it to the dynamics you seemed to describe.

Answer: We made this part more concise.

"Insects tend to congregate at the updraft zones of rolls and they can be seen as clear air echoes by weather radars."

The point is that the weather radar can be used as an effective tool in detecting rolls, since insects are usually present in the air during the summer season.

L 225 : induced not induced

Answer: Fixed.

Figure 8 : These two figures are pretty interesting but I think that you need the reader to understand what you show. So here there are apparently 2 event types : One regional and one induced by roll vortices. Looking at Figure 5b, I see several zones associated with high N3-20. One in the 4 first km north to SMEAR II mast and the second one is further north (12km). According to wind speed direction the one located further north

[Figure]

did not cross the site measurement. So my question is how could you separate the Roll vortices induced NPF from the regional one given the fact that both are located in the same zone ? If you used only the mean geometrical diameter, could you please justify why this is relevant ???

Answer: We know that the high N3-20 zone 4 km north of SMEAR II moved over the station and over the zeppelin's measurement area from south-west to north-east, which is perpendicular to the mean wind direction. This is illustrated in Figure 3 where the location of each concentrated particle stripe observation is put onto a map. The dot size for the zeppelin measurements is proportional to the altitude.

In addition, analysis of wind components measured from the top of the 125 m mast confirms that the rolls were moving in the same direction and at a rate consistent with the RI-NPF observation in Figure 8 B (Figure 4). The rolls were also observed in the weather radar image as parallel lines of higher reflectance (Figure 5).

In Figure 4, vz, v∥ and v≥ refer to the vertical wind component, the wind component along the rolls (direction checked from weather radar) and the wind component perpendicular to rolls (positive direction to the left side of the parallel wind component). All components have a low-frequency peak at 4e-4 Hz and the phase differences are consistent with rolls moving to the north-east of the station (see the methods section on detection of roll vortices). 4e-4 Hz is consistent with one roll moving over the station in about 20 minutes.

From the airplane the high N3-20 zone 4 km north of SMEAR II was observed around 10:00 AM, just when it had moved over the field station. The particle region 12 km north of Hyytiälä was observed at the end of the flight (around 11:30 AM). Probably the RI-NPF event moved further north-east with the rolls during the measurement, or it could be that this is a new RI-NPF occurring in an adjacent roll or rolls that previously did not extend all the way to the measurement area. The roll vortices are not perfectly straight continuous structures and undergo change over time.

In Figure 8 the mean geometric diameters were fitted over the growing particle mode and then we chose the time periods when we were measuring the concentrated particle stripes (that is we were measuring the RI-NPF, where the regional NPF was enhanced) and when we were not measuring them (we were only measuring the regional NPF).

[Figure]

**Fig. 1.** An illustration of how a difference in the direction of the mean wind and the roll axis causes the rolls (and roll-induced NPF) to move over a stationary point perpendicular to the mean wind.

Fig. 2.

wind direction (deg)

wind speed (m/s)

hour of the day

2007-04-16
2007-06-10
2011-06-02
2012-03-28
2012-04-05
2012-04-30
2013-03-28
2013-05-11
2013-05-23
2014-03-27
2014-04-02
2015-09-13
2017-04-24

[Figure]

Fig. 3.

[Figure]

$$\phi(v_{\perp}) - \phi(v_{\parallel}) = 71°$$
$$\phi(v_{\perp}) - \phi(v_z) = 293°$$
$$\phi(v_{\parallel}) - \phi(v_z) = 222°$$

Fig. 4.

Reflectivity in dBZ

| | |
|---|---|
| 12 | >14 |
| 10 | 12 |
| 8 | 10 |
| 6 | 8 |
| 4 | 6 |
| 2 | 4 |
| 0 | 2 |
| -2 | 0 |
| -4 | -2 |
| -6 | -4 |
| -8 | -6 |
| -10 | -8 |
| -12 | -10 |
| -14 | -12 |
| -16 | -14 |
| -18 | -16 |

**reflectivity and radial velocity
1.5 degrees elevation angle
11:45 EET**

Mean Velocity in m/s

| | |
|---|---|
| 6.6 | >7.6 |
| 5.7 | 6.6 |
| 4.7 | 5.7 |
| 3.8 | 4.7 |
| 2.8 | 3.8 |
| 1.9 | 2.8 |
| 0.9 | 1.9 |
| 0.0 | 0.9 |
| -0.9 | 0.0 |
| -1.9 | -0.9 |
| -2.8 | -1.9 |
| -3.8 | -2.8 |
| -4.7 | -3.8 |
| -5.7 | -4.7 |
| -6.6 | -5.7 |
| <-7.6 | -6.6 |

**11:00 EET**  **11:15 EET**  **11:30 EET**  **11:45 EET**

50 km

✕ SMEAR II

**Ikaalinen radar 2013-05-08
unfiltered reflectivity
0.3 degrees elevation angle**

Lat.-long. grid and lakes shown on map overlay

50 km

**Fig. 5.**

---

## Referee Comment (RC2) · Anonymous Referee #3 · 15 Jun 2020

General Comments

This study presents evidence from field data of the formation of aerosol particles from volatile organic compounds (New Particle Formation, NPF) due to the transport of boreal forest air to the upper regions of the atmospheric boundary layer by the convective boundary layer rolls. This is a relevant topic that deserves to be studied and understood, since it can have direct impact on the estimation and modeling of aerosols in the atmosphere, which are relevant for air quality, weather and climate. This study presents a dataset that shows clear evidence of the relationship between convective rolls and NPF. However, the manuscript needs some improvement in terms of the scientific writing. Due to its relevance, I suggest (1) improvements to the scientific presentation of the study, and (2) some additional analysis and discussion that can help future studies

on the development of better measurements and models for this phenomenon.

- Introduction: it is too short and some important information is lacking. For example, it needs more details on what is NPF (how it is defined, range of particle sizes of interest, where it comes from), why it is important (where it is used, where it is not used but should be used) and what are the mechanisms in which ABL dynamics might influence NPF. It would be important to describe in details what is already known about the relationship between NPF and convective rolls, what is not known (or never observed in field data), and what will be investigated here exactly. Why convective rolls, but not convective conditions in general? With this information the reader should be convinced about the relevance of this study. Right now this description (and consequently the motivation) of the study is superficial, only someone in the field will recognize its importance. It is important to convince the general audience as well. Some interesting information that should be in the intro is mentioned in the Conclusion section and in the caption of Figure 10.

- Methods: the section already starts with "Zeppelin measurements", without introducing the reader with the big picture of the methods of the study. It would be useful to start with a overall description (type of data, location, overall goal with each type of data, etc). After situating the reader, then go to the details. All the details needed to reproduce the analysis should be given. Some information is described in the results section (or in the caption of figures), some is missing (see details below). I'm very confused about the different particle size ranges mentioned in different moments of the manuscript. It there a range of interest?

- Results: these results are very interesting, but they are too focused on the measurements of particles, but not on the atmospheric conditions. Maybe the gas and meteorological data at the surface could be used to provide quantitative information about the roll-induced NPF? It would be interesting to characterize the roll

days with their micrometeorological variables, and to try to better identify the differences between the days with and without NPF. If this is not possible, it should be addressed in the manuscript, with a discussion of what should be done in future field studies in order to provide better quantitative data that can be used to model this phenomenon.

Specific Comments

- l. 25: "the small clusters and particles originating from these bursts grow in size similar to particles typically ascribed to regional scale atmospheric NPF". The difference between regional scale NPF and rolls induced NPF should be made clearer.

- l. 40: "In observational studies enhanced nucleation mode particle concentrations have been observed in turbulent layers in the lower atmosphere. For example inside the residual layer (Wehner et al., 2010) and in the inversion capping a shallow mixed layer (Platis et al., 2015; Siebert et al., 2004)." It is not clear how these two layers would favor the development of NPF, compared to other ABL conditions.

- l. 40: what is "nucleation mode particle"?

- l. 43: "Other airborne measurements have found significant horizontal and vertical variability in the number concentration of nucleation mode particles within the BL." Can you expand on that? What level of variability? Anything measured within the ABL has variability, what makes this one worth pursuing?

- l. 49: "Convection in the planetary BL often organizes into counter-rotating horizontal roll vortices or rolls that extend to the top of the boundary layer". What is the horizontal and time scales of these rolls? How can they be identified by
micrometeorological variables? This is relevant to evaluate if the measurements are appropriate. Why this specific type of convection is more relevant for NPF than others?

- l. 55: "and the overall effect of rolls on aerosol particle formation is unknown". Is it completely unknown? Can you be more specific on what is known, what is unknown? You have cited papers that discuss this.

- l. 68: "We used the positively charged particles and the data was averaged to 4 min time resolution". Why? Is that equivalent to the total concentration of particles?

- l. 72: "The data was corrected for diffusional losses in the one meter long, 37 mm inner diameter, inlet tube and converted to standard conditions (293.15 K and 1 atm)." How? Can you provide at least a reference, so that someone could reproduce what was done exactly?

- Sec 2.1: it is not clear after this section if the zeppelin data is only profiles or if there are measurements fixed at a given height.

- l. 89: "Particle number concentration in the 3-20 nm range was calculated by subtracting the total particle number concentration measured by the Scanning Mobility Particle Sizer (SMPS) from the number concentration measured by the Ultrafine Condensation Particle Counter (UCPC)." Not clear what that means. Why are you interested in this range only? The SMPS is mentioned in Table 1 as measuring between 10-400 nm. No information about UCPC is given. This description is not clear.

- l. 91: "The SMPS starts to lose accuracy in terms of spatial distribution of the aerosol particles due to its 2 min averaging period when the horizontal scale becomes less than 4 km." How does that apply to your study? Is this scale

comparable to the phenomenon that you are investigating? Is this relevant? What about the other instruments used?

- l. 93: "A turbulence probe, capable of measuring the 3d wind vector, was only installed at the end of the 2015 campaign." This sentence is completely lost here. What is this going to be used for? And how? Any details on this instrument? Measurement frequency, probe model, post processing?

- Sec 2.2: it is also not clear after this section if the airplane data used is only profiles or if there are measurements fixed at a given height.

- l. 122: "The CPC had a 10 nm cutoff size" what is the measurement range? what is CPC?

- l. 132: what is Aitken mode?

- Sec 2.4: Is there an exact quantitative criteria for NPF days, or was it selected by inspection only?

- l. 147: What is the time interval used? What size ranges are used? How is the coagulation sink obtained? It is important to provide all information from the data to the results presented.

- Sec. 2.8: I did not see the use of the ABL height in the results section.

- l. 193: "Figure 2 shows a frequent observation in the measurement data:" which data?

- Results section: why is the particle range size different in different analysis (for example figs 4 and 5, or between conditions (i) and (ii))

- l. 208-216: this paragraph should be in the Methods section.

- l. 228-229: which statistic test was performed? All information necessary to reproduce your results should be given.

- l. 229-232: can you verify in the data what micrometeorology conditions characterize NPF and non-NPF days?

- l. 235: "This timescale is associated with mixing throughout the convective BL" did you calculate it? Compared with references?

- l. 238-244: instead of Table 3, it would be useful to show plots related to the estimation of GR. Also, what is the particle range size of your GR estimate?

- Figure 8: "and power spectra of the wind components from the station's mast showed that the rolls were moving over the site" this would be interesting to see, maybe it could be added to this figure as a third panel?

- The analysis in Figure 9 is not clear to me. It needs to be better explained in the methods and results section, not only in a figure caption. All the details needed to reproduce your results should be presented.

- Figure 10 is more appropriate for the introduction than conclusion. A good description of the physical process that motivates this study is in the caption of the figure, and it would be important for the reader to know about these things since the beginning.

Technical Corrections

- l. 40: "In observational studies, enhanced" (add the comma)

- l. 52: "(Buzorius et al., 2001; Nilsson et al., 2001)" you don't have to cite the same thing twice on the same sentence.

- l. 55: "However direct observations (...)", rephrase.

- l. 67: what is "mobility diameter"?

- l. 79: "while the airspeed was kept at 20 m/s" not clear what that means

- Table captions: remove the word "Explanations:"

- l. 85: Table 1 also mentions the Zepelin data, why is it mentioned only in the Airplane section?

- l. 96: "such that the aircraft was either descending, ascending or staying level", maybe rephrase as "measurements performed during descending, ascending..."

- l. 98: "The measurement airspeed was 36 m/s", again, not clear.

- It goes from section 2.4 to section 2.8

- l. 143: why no equation number?

- l. 221: "station. Whereas" change to comma

- I don't think Table 2 is necessary, the statistics are sufficient.

- l. 225: "roll-indcued"

- Table 3: as Table 2, I don't think this is necessary. It should be presented the statistics, but the information for each individual day is not necessary for the understanding of the study. If you decide to keep these tables, maybe put them in an appendix or supplemental material.

- l. 265: equation number

---

## Author Comment (AC2) · 7 Jul 2020

We thank the referee for the constructive comments, please find our responses below.

General Comments

This study presents evidence from field data of the formation of aerosol particles from volatile organic compounds (New Particle Formation, NPF) due to the transport of boreal forest air to the upper regions of the atmospheric boundary layer by the convective boundary layer rolls. This is a relevant topic that deserves to be studied and understood, since it can have direct impact on the estimation and modeling of aerosols in the atmosphere, which are relevant for air quality, weather and climate. This study presents a dataset that shows clear evidence of the relationship between convective rolls and
NPF. However, the manuscript needs some improvement in terms of the scientific writing. Due to its relevance, I suggest (1) improvements to the scientific presentation of the study, and (2) some additional analysis and discussion that can help future studies on the development of better measurements and models for this phenomenon.

Introduction: it is too short and some important information is lacking. For ex- ample, it needs more details on what is NPF (how it is defined, range of particle sizes of interest, where it comes from), why it is important (where it is used, where it is not used but should be used) and what are the mechanisms in which ABL dynamics might influence NPF. It would be important to describe in details what is already known about the relationship between NPF and convective rolls, what is not known (or never observed in field data), and what will be investigated here exactly. Why convective rolls, but not convective conditions in general? With this information the reader should be convinced about the relevance of this study. Right now this description (and consequently the motivation) of the study is su- perficial, only someone in the field will recognize its importance. It is important to convince the general audience as well. Some interesting information that should be in the intro is mentioned in the Conclusion section and in the caption of Figure 10.

Answer: we extended the first paragraph in the introduction to give more details on NPF. We moved the explanation on how roll vortices could induce NPF above the boreal forest from the conclusions to the introduction along with Figure 10. We made the the rest of the introduction mode detailed by following to the comments below.

Methods: the section already starts with "Zeppelin measurements", without introducing the reader with the big picture of the methods of the study. It would be useful to start with a overall description (type of data, location, overall goal with each type of data, etc). After situating the reader, then go to the details. All the details needed to reproduce the analysis should be given. Some information is described in the results section (or in the caption of figures), some is missing (see details below). I'm very confused about the different particle size ranges mentioned in different moments of the manuscript. It

there a range of interest?

Answer: we added in the beginning a paragraph giving an overview of the measurements. In the detailed sections we took the below comments into account and rephrased or added text. In the conditions for roll-induced NPF we were looking at sub-20 nm particles, since this data was readily available from all measurement platforms.

Results: these results are very interesting, but they are too focused on the measurements of particles, but not on the atmospheric conditions. Maybe the gas and meteorological data at the surface could be used to provide quantitative information about the roll-induced NPF? It would be interesting to characterize the roll days with their micrometeorological variables, and to try to better identify the differences between the days with and without NPF. If this is not possible, it should be addressed in the manuscript, with a discussion of what should be done in future field studies in order to provide better quantitative data that can be used to model this phenomenon.

Answer: we agree that the analysis could be expanded. For example developing more comprehensive methods to measure the phenomenon and studying the cluster composition during roll-induced NPF. However we find this further analysis is beyond the scope of this study.

We added to the conclusions: "In order to fully understand roll-induced NPF, better measurement and analysis methods need to be developed. For example measuring the fluxes of sub-10 nm particles and doing airborne flux measurements. More measurements with a turbulence probe on board need to be performed. It would also be interesting to study the cluster composition during roll-induced NPF."

Specific Comments

l. 25: "the small clusters and particles originating from these bursts grow in size similar to particles typically ascribed to regional scale atmospheric NPF". The difference

between regional scale NPF and rolls induced NPF should be made clearer.

Answer: we rephrased the text to show the difference between roll-induced and regional scale NPF more clearly:

"the small clusters and particles originating from these localized bursts grow in size similar to particles typically ascribed to atmospheric NPF that occurs almost homogeneously at a regional scale."

l. 40: "In observational studies enhanced nucleation mode particle concentrations have been observed in turbulent layers in the lower atmosphere. For example inside the residual layer (Wehner et al., 2010) and in the inversion capping a shallow mixed layer (Platis et al., 2015; Siebert et al., 2004)." It is not clear how these two layers would favor the development of NPF, compared to other ABL conditions.

Answer: we added text explaining why BL dynamics can be important for NPF:

"Numerical studies have shown that fluctuations in ambient temperature and relative humidity, caused by for example small-scale turbulence, large eddies such as roll vortices (Easter and Peters, 1994), or mixing over a temperature inversion (Nilsson and Kulmala, 1998) can lead to significant enhancements in new particle formation rate compared to only mean conditions. This is because the formation rate has a non-linear dependence on temperature and the gas-phase concentrations of the precursor vapors. Therefore, fluctuations in these variables, as opposed to mean conditions where the fluctuations are averaged out, can have a net enhancing effect on the source strength of aerosol particles by NPF."

Now it should be more clear why turbulence would favor NPF in these layers. We also edited the text a bit:

"In observational studies, increased nucleation mode particle concentrations have been measured in atmospheric layers where turbulent fluctuations were enhanced. For example in turbulent layers inside the residual layer (Wehner et al., 2010) and in

the inversion capping a shallow mixed layer (Platis et al., 2015; Siebert et al., 2004)."

l. 40: what is "nucleation mode particle"?

Answer: we added "sub-25 nm" to the text where we introduce NPF.

l. 43: "Other airborne measurements have found significant horizontal and vertical variability in the number concentration of nucleation mode particles within the BL." Can you expand on that? What level of variability? Anything measured within the ABL has variability, what makes this one worth pursuing?

Answer: we added text about the degree of variation that can be found

"Other airborne measurements have found that during NPF the number concentration of nucleation mode particles shows considerable, up to an order of magnitude, variation within the BL"

l. 49: "Convection in the planetary BL often organizes into counter-rotating horizontal roll vortices or rolls that extend to the top of the boundary layer". What is the horizontal and time scales of these rolls? How can they be identified by micrometeorological variables? This is relevant to evaluate if the measurements are appropriate. Why this specific type of convection is more relevant for NPF than others?

Answer: we added Figure 1 that shows 3d view of roll circulation with labels that explain the scale. The methods to identify roll vortices in the BL are outlined in the methods section. We also moved the explanation and the associated figure of the concept behind roll-induced NPF from the conclusions to the introduction.

l. 55: "and the overall effect of rolls on aerosol particle formation is unknown". Is it completely unknown? Can you be more specific on what is known, what is unknown? You have cited papers that discuss this.

Answer: we rephrased the text to be more specific

"However direct observations of the effects of roll vortices on NPF are lacking."

l. 68: "We used the positively charged particles and the data was averaged to 4 min time resolution". Why? Is that equivalent to the total concentration of particles?

Answer: we modified the text to read

"We used the total particle data from the positive polarity of the instrument."

In our case both polarities looked roughly the same in terms of data quality so the choice was more or less arbitrary, but in any case one should perform the analysis on one polarity only so that the data is most comparable.

l. 72: "The data was corrected for diffusional losses in the one meter long, 37 mm inner diameter, inlet tube and converted to standard conditions (293.15 K and 1 atm)." How? Can you provide at least a reference, so that someone could reproduce what was done exactly?

Answer: we added a reference to the diffusion loss calculation

"Gormley, P. G. and Kennedy, M.: Diffusion from a Stream Flowing through a Cylindrical Tube, Proc. R. Ir. Acad. Sect. Math. Phys. Sci., 52, 163–169, 1948." We also added the equation for the conversion to standard conditions.

Sec 2.1: it is not clear after this section if the zeppelin data is only profiles or if there are measurements fixed at a given height.

Answer: we made the text more specific:

"The zeppelin measurements consisted of consecutive profiles. Each profile was a slow and even ascend (∼25 min) from ∼100 m up to ∼1 km above ground followed by a fast descend (∼5 min) while the speed relative to the surrounding air (airspeed) was kept at ∼20 m/s."

l. 89: "Particle number concentration in the 3-20 nm range was calculated by subtracting the total particle number concentration measured by the Scanning Mobility Particle Sizer (SMPS) from the number concentration measured by the Ultrafine Condensation

Particle Counter (UCPC)." Not clear what that means. Why are you interested in this range only? The SMPS is mentioned in Table 1 as measuring between 10-400 nm. No information about UCPC is given. This description is not clear.

Answer: we modified the text

"We used the particle number concentration in the 3-20 nm size range as an indication of particles that likely originated from NPF. The 3-20 nm particle number concentration was calculated by subtracting the total particle number concentration measured by the Scanning Mobility Particle Sizer (SMPS) in the size range 20-400 nm from the number concentration measured by the Ultrafine Condensation Particle Counter (UCPC). We skipped the smallest size bins of the SMPS because they were in some cases noisy."

We also added UCPC in the Table 1.

l. 91: "The SMPS starts to lose accuracy in terms of spatial distribution of the aerosol particles due to its 2 min averaging period when the horizontal scale becomes less than 4 km." How does that apply to your study? Is this scale comparable to the phenomenon that you are investigating? Is this relevant? What about the other instruments used?

Answer: we decided to leave this part out because of the following reasons:

During the roll-induced NPF observations the number concentration from the UCPC was elevated during a large part of at least one SMPS scan. In these cases the SMPS total number concentration did not increase at all (the particles were below the detection limit) or the number concentration was momentarily (one or more SMPS scans) increased in the smallest size bins (10-20 nm) of the SMPS. An example is presented in Figure 2 where purple arrows show the times when the airplane flew through a roll-induced NPF.

In light of this we would say that the calculated 3-20 nm number concentration was in our cases a reliable indication that the number concentration was increased in the 3-20 size range. Therefore mentioning this limitation here is not relevant and can lead to

confusion.

l. 93: "A turbulence probe, capable of measuring the 3d wind vector, was only installed at the end of the 2015 campaign." This sentence is completely lost here. What is this going to be used for? And how? Any details on this instrument? Measurement frequency, probe model, post processing?

Answer: we added the following paragraph:

"In order to detect roll vortices on board the airplane we installed a turbulence probe (Aventech Research, AIMMS-20) at the end of the 2015 campaign. The AIMMS-20 was capable of measuring the the 3d wind vector at 20 Hz, but for the analysis we averaged the data to 1 s."

Sec 2.2: it is also not clear after this section if the airplane data used is only profiles or if there are measurements fixed at a given height.

Answer: we wrote the following:

"Typical measurement tracks consisted of ~30 km long flight segments flown roughly perpendicular to the mean wind direction over the same area while doing a single vertical profile from 100 m to 3000 m above ground. The ascend and descend speeds were on average ~1 m/s."

l. 122: "The CPC had a 10 nm cutoff size" what is the measurement range? what is CPC?

Answer: the CPC measured all particles above 10 nm. We rephrased the text:

"The system measuring the vertical particle flux used an ultrasonic 3d anemometer combined with a condensation particle counter (CPC) at 23 m above ground. The CPC had a 10-nm cutoff size."

l. 132: what is Aitken mode?

Answer: we added "Aitken mode (25-100 nm)"

Sec 2.4: Is there an exact quantitative criteria for NPF days, or was it selected by inspection only?

Answer: this is done by inspection, we added this to the text.

l. 147: What is the time interval used? What size ranges are used? How is the coagulation sink obtained? It is important to provide all information from the data to the results presented.

Answer: we added the following:

"We calculated the CoagSd from the DMPS data and for the number concentrations we used the NAIS data, so that the final time resolution of the formation rate was 4 min. The size ranges used from the NAIS data were 3-6 nm for J3 and 10-20 nm for J10."

Sec. 2.8: I did not see the use of the ABL height in the results section.

Answer: in roll-induced NPF condition (i) we specify that the concentrated longitudinal sub-20 nm particle zone should be inside the BL and this is where we checked the BL height.

l. 193: "Figure 2 shows a frequent observation in the measurement data:" which data?

Answer: rephrased to "Figure 2 shows a frequent observation in the ground-based aerosol particle measurements"

Results section: why is the particle range size different in different analysis (for example figs 4 and 5, or between conditions (i) and (ii))

Answer: in the roll-induced NPF conditions (i) and (ii) we are looking at sub-20 nm particles.

We changed the size range in Figure 4 to be 3-20 nm instead of >1.5 nm in order to be consistent. In Fig. 5 the size range is also 3-20 nm.

In Figure 3 the SMPS stopped working in the middle of the flight, which is why we are only showing data from the UCPC (>3 nm). However, we know from the simultaneous ground-based observations that the observed particles were sub-20 nm.

l. 208-216: this paragraph should be in the Methods section.

Answer: we think it is necessary to explain the case study before defining the two roll-induced NPF conditions. Otherwise it is very difficult for the reader to understand why we define the conditions the way we do.

l. 228-229: which statistic test was performed? All information necessary to reproduce your results should be given.

Answer: the statistical test was Fisher's exact test. It is mentioned in the text.

l. 229-232: can you verify in the data what micrometeorology conditions characterize NPF and non-NPF days?

Answer: NPF events generally occur on sunny days with a lot of atmospheric mixing. Our data does agree with this (see Figure 3). However this figure adds little to understanding roll-induced NPF so we chose to leave it out of the manuscript.

l. 235: "This timescale is associated with mixing throughout the convective BL" did you calculate it? Compared with references?

Answer: we changed this sentence to be more specific to rolls: "This timescale is similar to the period of a typical roll vortex". A reference (Easter and Peters, 1994) was given in the introduction. This time scale is also similar to the mixing throughout the BL since the rolls circulate the air throughout the depth of the BL.

l. 238-244: instead of Table 3, it would be useful to show plots related to the estimation of GR. Also, what is the particle range size of your GR estimate?

Answer: we see that a good example of growing roll-induced NPF particles is already shown in Figure 8 where mean mode diameters are fitted to multiple subsequent rollinduced NPF observations and over time they show a growth trend. We added fit lines and the resulting GRs to the figure also. For the GR estimate the median lower size was 7.5 nm and the median upper size was 15 nm. We added this information to the text.

Figure 8: "and power spectra of the wind components from the station's mast showed that the rolls were moving over the site" this would be interesting to see, maybe it could be added to this figure as a third panel?

Answer: the power spectra, along with two other supporting figures showing the movement of the roll-induced NPF and the rolls in the weather radar image, can be seen in our reply to Referee #1 (Fig. 4, explanation of the figure is in the answers). We find that the figure is quite technical and would not add significant extra value.

We made the description in the caption more precise:

"The roll-induced NPF event was moving over the measurement area from southwest to northeast. Weather radar observations showed that rolls were present over the measurement site and power spectra of the wind components from the station's mast showed that the rolls were moving over the site at the same rate (one roll in $\sim$20 min), and in the same direction as the roll-induced NPF."

The analysis in Figure 9 is not clear to me. It needs to be better explained in the methods and results section, not only in a figure caption. All the details needed to reproduce your results should be presented.

Answer: we added a detailed explanation to the text, see our answer to the first referee.

Figure 10 is more appropriate for the introduction than conclusion. A good de- scription of the physical process that motivates this study is in the caption of the figure, and it would be important for the reader to know about these things since the beginning.

Answer: we moved the figure to the introduction.

Technical Corrections

l. 40: "In observational studies, enhanced" (add the comma)

Answer: fixed

l. 52: "(Buzorius et al., 2001; Nilsson et al., 2001)" you don't have to cite the same thing twice on the same sentence.

Answer: fixed

l. 55: "However direct observations (...)", rephrase.

Answer: we rephrased this to "However direct observations of the effects of roll vortices on NPF are lacking."

l. 67: what is "mobility diameter"?

Answer: the SMPS, DMPS and NAIS diameters are electrical mobility equivalent diameters. We decided to refer to simply diameters throughout the text to avoid confusion.

l. 79: "while the airspeed was kept at 20 m/s" not clear what that means

Answer: we modified the text to read "the speed relative to the surrounding air (airspeed) was kept at ∼20 m/s."

Table captions: remove the word "Explanations:"

Answer: fixed

l. 85: Table 1 also mentions the Zepelin data, why is it mentioned only in the Airplane section?

Answer: we moved the sentence to the overview in the beginning of the methods section.

l. 96: "such that the aircraft was either descending, ascending or staying level", maybe rephrase as "measurements performed during descending, ascending..."

Answer: rephrased

"Typical measurement tracks consisted of ~30 km long flight segments flown roughly perpendicular to the mean wind direction over the same area while doing a single vertical profile from 100 m to 3000 m above ground (Figure 3). The ascend and descend speeds were ~1 m/s."

l. 98: "The measurement airspeed was 36 m/s", again, not clear.

Answer: the airspeed is now explained in the Zeppelin section, so this should be clear.

It goes from section 2.4 to section 2.8

Answer: fixed

l. 143: why no equation number?

Answer: equation number added

l. 221: "station. Whereas" change to comma

Answer: changed

I don't think Table 2 is necessary, the statistics are sufficient.

Answer: we agree that the detailed information presented is not necessary, so we removed the table and the references to it from the text.

l. 225: "roll-inducued"

Answer: fixed

Table 3: as Table 2, I don't think this is necessary. It should be presented the statistics, but the information for each individual day is not necessary for the understanding of the study. If you decide to keep these tables, maybe put them in an appendix or supplemental material.

Answer: we agree and decided to remove Table 3 from the text. The important statistics

(average growth and formation rates) can be found in the text and the important figure derived from Table 3 is Figure 9.

l. 265: equation number

Answer: equation number added
* * *
[Figure]

Mean wind

Roll vortex

Boundary layer
1-2 km

Surface

1-5 km

**Fig. 1.**

![Figure 2: Particle size distribution diagram for 2014-04-02. X-axis shows Hour of the day from 9.75 to 11.50. Left Y-axis shows Diameter in meters (10^-8 to 10^-7). Right Y-axis shows Number concentration > 3 nm (cm-3) from 0 to 35000. Color scale shows dN/dlogDp in cm^-3 from 10^1 to 10^5.]

**Fig. 2.**

Fig. 3.

---

## Author Response (AR2)

Technical suggestion:
- l. 75: Figure 2 without parenthesis

Answer: removed the parenthesis

- l. 77: "However," (add comma), or modify this sentence so it does not start with "However"

Answer: removed the word "However" from the sentence

- l. 100: "During the measurement," (add comma)

Answer: added a comma

- l. 295: GR sometimes in italic, sometimes not

Answer: changed to not italic

- l. 305: maybe this should not be a new paragraph

Answer: combined the paragraphs

- l. 305: "We assumed that the roll-induced NPF GRs were normally distributed with mean equal to the median GR and standard deviation given by the magnitude of the IQR" I'm confused by this choice of parameter estimation. If the data follows a normal distribution, why not use the sample mean and standard deviation as estimators of the distribution? Why mean = median, and standard deviation = IQR? Is that equivalent?

Answer: we wanted to minimize the effect of outliers on the results and chose median and IQR as the measures of central tendency and spread. If we assume that the binned data follows a normal distribution then the median and the mean should be the same. However, it is true that for normal distribution IQR ~ 1.35*SD, where SD is the standard deviation. Therefore in order to estimate the SD using the IQR we should use SD = IQR/1.35.

While going through the code we noticed that the GR was calculated by using the time in the middle of the RI-NPF observation. However, the mean particle size for each RI-NPF was reported for the beginning of the observation. Therefore we should also use the time that marks the beginning of the observation. We fixed this, and as expected the value for the GR did not change very much (median GR: 1.9 → 1.8 nm/h). In any case we corrected the value in the text and added a sentence mentioning this choice.

We also noticed a bug in the random sampling python script that calculated the formation rates and their uncertainties. We fixed the bug and repeated the analysis using SD = IQR/1.35. Not surprisingly the values changed somewhat (median $J_3$: 2.4 cm$^{-3}$s$^{-1}$ → 1.9 cm$^{-3}$s$^{-1}$, the median $J_{10}$ stayed at 0.4 cm$^{-3}$s$^{-1}$). However, we believe that this does not change the conclusions presented in the paper. We made the description of the random sampling method more detailed in the text.

We also noticed one other possible source of confusion in the text. In the random sampling method we were interested in collecting statistics on the peak formation rates. The time when the peak occurs depends on the sample, therefore the peak formation rate in the median formation rate time series (shown in Figure 12) does not correspond to the median peak formation rate of all samples. Because of this we decided to remove the median formation rate time series from Figure 12.

[revised manuscript text omitted]